# Seasonal Abundance, Density and Distribution of Cetaceans in the Bulgarian Black Sea Shelf in 2017

Dimitar Popov [1,2,*], Marina Panayotova [3], Radoslava Bekova [3], Hristo Dimitrov [2] and Galina Meshkova [1]

[1] Green Balkans NGO, 1A Skopie Str., 4013 Plovdiv, Bulgaria
[2] Zoology Department, Biology Faculty, University of Plovdiv, 2 Todor Samodumov Str., 4000 Plovdiv, Bulgaria
[3] Institute of Oceanology–Bulgarian Academy of Sciences, 40 Parvi May Str., P.O. Box 152, 9000 Varna, Bulgaria
[*] Correspondence: dpopov@greenbalkans.org; Tel.: +359-885108712

**Abstract:** Black Sea cetaceans are isolated and nominated as endemic subspecies listed in the IUCN Red List of Endangered species: the harbour porpoise and bottlenose dolphin as Endangered (EN) and the common dolphin as Vulnerable (VU). Studies of their distribution and abundance are scarce but obligatory for assessment of their conservation status. Being highly mobile apex predators entails large variations in spatial and temporal distribution. Two vessel line–transect distance sampling surveys were conducted in 2017 in the Bulgarian shelf with the aim to estimate the density, abundance and distribution during spring and autumn. Results have revealed a shift from the coastal to offshore shelf of harbour porpoises with marked southern movement. Density of porpoises varied from 1.423 ind./km$^2$ (CV = 25.4%) in spring to 0.576 ind./km$^2$ (CV = 43.43%) in autumn. The density of common dolphins was also decreasing, from 0.391 ind./km$^2$ (CV = 36.84%) to 0.088 ind./km$^2$ (CV = 42.13%), which was more significant in offshore (0.031 ind./km$^2$, CV = 58.04%) than in the coastal shelf (0.138 ind./km$^2$, CV = 48.59%). Bottlenose dolphins had almost constant density in both seasons in the coastal shelf: 0.211 ind./km$^2$ (CV = 52.15%) and 0.187 ind./km$^2$ (CV = 52.13%) but a very low density in the offshore shelf in autumn: 0.042 ind./km$^2$ (CV = 71.07%). The importance of existing NATURA 2000 sites for the harbour porpoise (BG0000621 Shabla-Ezerets, BG0000573 Kompleks Kaliaka, BG0001001 Ropotamo and BG0001007 Strandzha) and bottlenose dolphin (BG0000621 Shabla-Ezerets, BG0000573 Kompleks Kaliaka, BG0001501 Emona and BG0001001 Ropotamo) were confirmed.

**Keywords:** Black Sea cetaceans; seasonal distribution; abundance; density; spatial distribution; Natura 2000

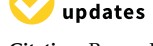



## 1. Introduction

The Black Sea is an isolated basin connected to the Mediterranean Sea only through the narrow Bosphorus Strait. Isolation and high inflow of nutrient-rich freshwater by some of the largest European rivers result in its low salinity of 14–18% and high eutrophication [1]. Waters below depth of 100–150 m are characterized by a high content of H$_2$S, defining it as the largest anoxic water body on Earth. All these features define it as a fragile and highly vulnerable ecosystem. Cetaceans inhabiting the Black Sea are isolated and nominated as endemic subspecies: the Black Sea harbour porpoise (*Phocoena phocoena relicta*, Abel, 1905), Black Sea bottlenose dolphin (*Tursiops truncatus ponticus* Barabash-Nikiforov, 1940) and Black Sea common dolphin (*Delphinus delphis ponticus* Barabash-Nikiforov, 1935). All three are listed in the IUCN Red List of Endangered species: the first two as Endangered (EN) [2,3] and the last as Vulnerable (VU) [4]. The intensive industrialized dolphin fishery until 1983 has caused large decline in populations of these top predators in the Black Sea. Numerous threats have limited the recovery of populations of these apex predators, with pollution, biological invasions, overfishing, and bycatch being amongst the most significant [5–8]. With accession to the European Union, Bulgaria has committed to its

policy in biodiversity conservation epitomized by the EU Habitats Directive 92/43/EEC and the EU Marine Strategy Framework Directive 2008/56/EC further to its prior role as party to the Agreement on the Conservation of Cetaceans of the Black Sea, Mediterranean Sea, and contiguous Atlantic area (ACCOBAMS). As a member state, it is obliged to monitor and ensure favourable conservation status and good environmental status (GES) of species protected by EU Directives, cetaceans being amongst them. Reliable abundance estimates are critical for the conservation of all cetaceans and is one of the components for assessment of their status together with habitat for the species, distribution range, demographic structure and future prospects defined by threats. Collecting information required to assess cetaceans' status is a complex process due to their high mobility, completely aquatic life cycle, large distances they travel, variation in seasonal and spatial distribution, slow reproduction, and diverse threats and impacts. Regular monitoring over continuous periods is required to be able to detect population trends and collect required data to review status and formulate relevant conservation measures.

Data on abundance, density and distribution of cetaceans in Bulgarian waters is scarce and limited. Reconnaissance flights have been used to guide dolphin hunting fleet [9] in the 1960s that provide some basic information on spatial distribution of concentrations, group sizes and seasonality. After moratorium on the dolphin hunting, the reduced populations have not recovered due to prey depletion in the 1980s and 1990s [10]. During that period cetaceans have remained out of focus of marine biodiversity research. It was not until the 21st century that some data from opportunistic [11,12] and dedicated surveys [13–16] in Bulgarian Black Sea waters was published.

The paper presents results from two vessel line–transect distance sampling surveys targeting to obtain abundance and density estimations of cetaceans in the Bulgarian shelf of the Black Sea. The first survey was conducted in June 2017 and the second in November–December the same year. Even though the first survey covered only internal and territorial waters while the second entire shelf to 100 m depth, results represent a good basis for comparison of densities and distribution during two seasons, spring and autumn.

## 2. Materials and Methods

### 2.1. Study Area

The study area included coastal, territorial and shelf waters up to 100 m depth. Territorial waters fully encompass coastal and partially shelf. The survey, conducted in June, covered only territorial waters that have total area of 6358.09 km$^2$. This stratum includes a variety of bottom substrates, including fine and coarse sands, silts and rocks. The coastline is varied with two larger bays near Varna and Burgas. The most prominent capes are Kaliakra, Emine and Maslen where constant currents occasionally provide suitable conditions for localized upwellings. Depth varies from 0–30 m inshore to 30–80 m offshore with the greatest depths along the coast being observed in the north while offshore in the south. The second survey, in November–December, covered entire shelf waters from the coast to 100 m isobath offshore with a total area of 12,090.09 km$^2$. Thus, the territorial and coastal waters have been fully surveyed twice while the remaining deeper part of the shelf only once.

### 2.2. Survey Design

Design for both surveys was made using Distance 6.0 package. On the basis of equal coverage probability criterion, optimal design and effort were determined to comply with logistical and environmental constraints for the completion of surveys. In both cases, lines followed an equally spaced zigzag shape in the east-west orientation, perpendicular to the coast, to assure sampling across potential density gradient (depth) with the aim of reducing encounter rate variability. The total length of the 13 pre-determined track lines for the June survey was 442 km (Figure 1). That design ensured 7% coverage of the study area. Due to the shape of the study area, a design produced by Distance was not a continuous zigzag track line and gaps existed. Though in this way, coverage probability was equal and bias in effort at areas where lines were connecting was avoided. Twelve transects for the autumn

survey had a total length of 684 km, ensuring 5.7% coverage, which were continuous zigzag (Figure 2) with the aim to use optimally available shorter daylight conditions in that time of the year.

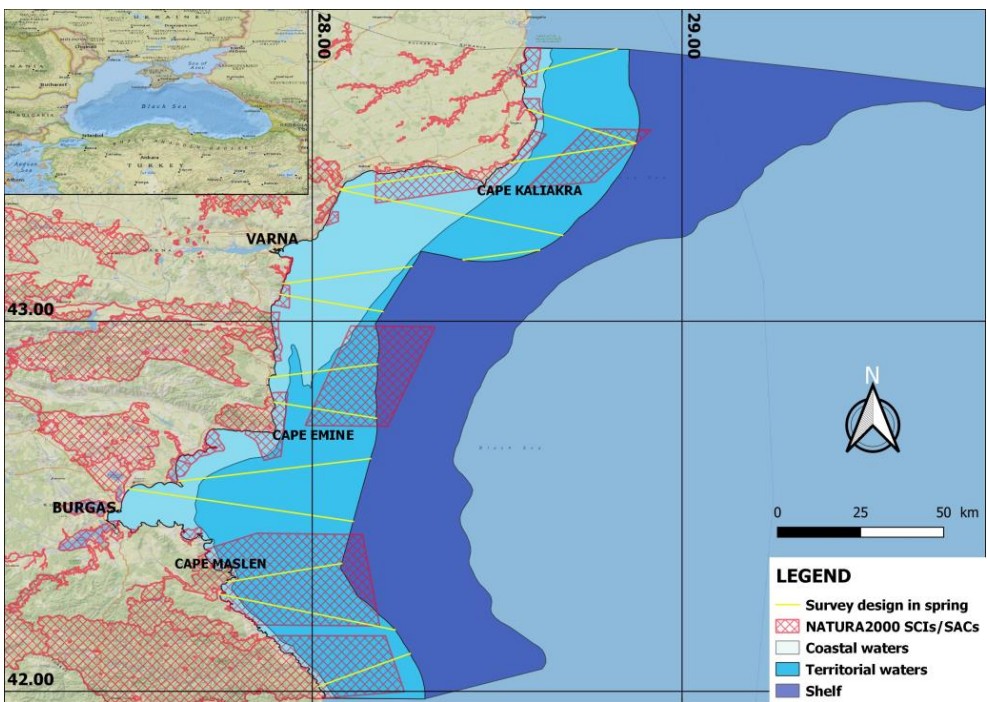

**Figure 1.** Design of the June survey.

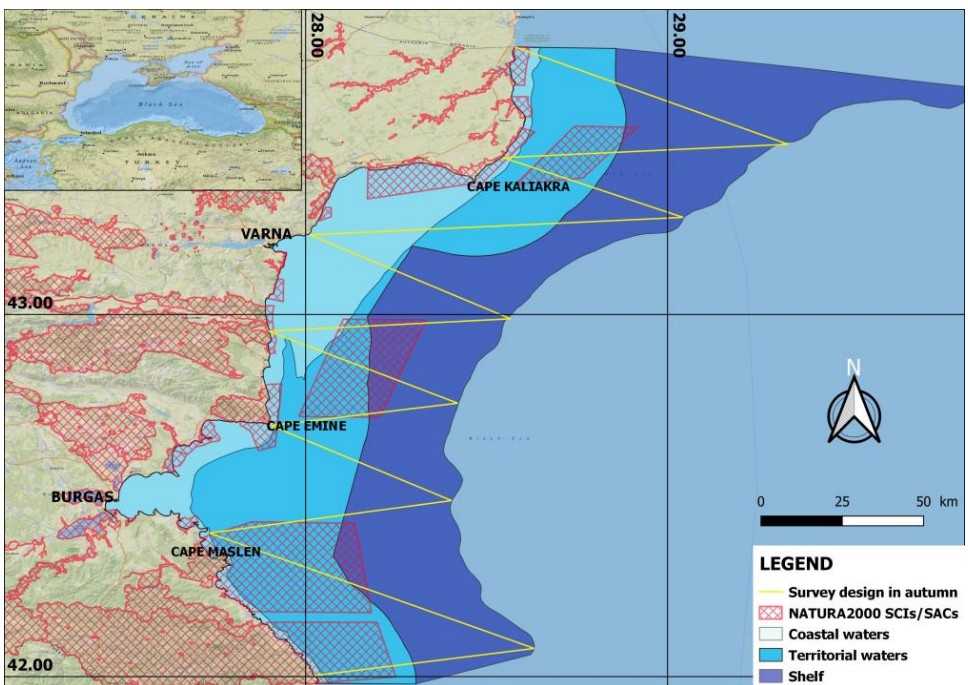

**Figure 2.** Design of the November–December survey.

### 2.3. Data Collection

During the surveys, the standard line–transect distance sampling [17,18] methodology was used deploying a single platform (two observers and one data recorder). Basically, the method includes traversing along predetermined transect lines and whenever an object of

interest (cetacean in our case) is observed, the distance to the object is recorded together with the radial angle (angle between object and transect line). The perpendicular distance of the object to the transect can then be calculated using simple trigonometry [18]. For each observation, the following information was recorded: date, time, distance, observer, radial angle, species, group size, behaviour and presence of calves. Data on environmental conditions, including sea state (in Beaufort scale), glare intensity, visibility and sightability, were also recorded at start of each transect, as well as the rotation of observers and when changes occurred. Surveys were conducted during appropriate weather conditions (sea state less than 4 by Beaufort scale and good visibility of more than 5 km). Observers scanned from abeam (90°) on their side to 10° on the opposite side with the naked eye and binoculars were used for identification of probable sightings and measuring the distance to these.

The survey in June used a 12.8 m motor-sailing yacht as platform that ensured a speed of 6.5 knots along the transects while on effort. Th platform's height was approx. 150 cm above sea level. Observers' eye height was individually measured seated and standing (214–318 cm) and were applied for the calculation of distances to observed cetaceans. The distance and radial angle of each observation were measured using PENTAX Marine 7 × 50 reticle binoculars with a built-in compass. Geographic coordinates of sightings and tracks of surveys were recorded by handheld GPS device (Garmin GPSMap 64st). Observers in the team have been trained during several pilot surveys conducted in 2015–2016 with the aim of gaining practical experience.

The survey in November–December used a 15.6 m fishing vessel as a platform that maintained a constant speed of 6–7 knots along transects. The platform's height was approx. 350 cm above sea level. Reticle binoculars BARSKA Deep Sea 7 × 50 with a built-in compass were used for measuring the distance and radial angle for each sighting. Observers in the autumn surveys had previous practical experience in distance sampling surveys and one of them was involved also in the spring survey.

*2.4. Data Analysis*

The abundance and density of animals and groups were estimated by analytical tools based on the detection probability functions for distance sampling [18] using specialized software Distance 7.4 [19]. Density ($\hat{D}$) is calculated using the following formula [20]:

$$\hat{D} = \frac{n}{2wL\hat{P}_a} = \frac{n}{2\hat{\mu}L} = \frac{n\hat{f}(0)}{2L}$$

where n is number of observed objects; L is sum of lengths of all transects; w is truncation distance from line; $\hat{P}_a$ is probability for detection; μ is effective strip (half-) width; and f(0) is probability density function. Abundance for animals occurring in clusters is calculated using following Horvitz-Thompson estimator [19]:

$$\hat{N} = \sum_{i=1}^{n} \frac{s_i}{\hat{P}_i}$$

where n is number of observations/sightings; $\hat{P}$ is the estimated inclusion probability for animal i and $S_i$ is the size of cluster i, i = 1, ... , n. Encounter rate was defined as a number of sightings (groups) per kilometre of effort. Population density was estimated as a number of individuals per square kilometre. Only sightings on effort along transects were used in the analysis. The group size estimation was made by the size bias regression method when regression was significant at an alpha level of 0.15; otherwise, the mean of observed group size was used. The minimum value of the Akaike Information Criterion or AIC [21] was used to choose between models of detection function. The AIC provides a relative measure of fit. The model with the smallest AIC provides, in some sense, the best fit to the data [19]. A difference of more than 2 for AIC values shows a better-fitting model. Results from performed analyses were compared also on basis of goodness-of-fit tests and we used

the Cramér–von Mises (CvM) family of tests that are performed by Distance 7.4 package. Selection of the most appropriate model of detection function, including a combination of key function and series expansions, was based on a procedure involving comparison of results of quantile–quantile plots, CvM goodness-of-fit tests where "p" had the highest values and AIC's lowest value. In addition, for models where the delta AIC was below 2, the model with a lower coefficient of variation was selected as the optimal. In this way, the most reliable density and abundance estimations were achieved.

Both conventional distance sampling (CDS) and multiple covariate distance sampling (MCDS) with species and sea state used as covariates were executed for data from June. Only CDS analysis was performed for data from the autumn survey as no data on environmental conditions was available. As the survey in autumn covered a larger study area, stratification was applied, dividing it into two strata: territorial waters of the shelf that fully overlap with the study area in June and the offshore shelf. Relative densities and encounter rates were calculated for separate strata together with global density and abundance for the entire shelf. Heatmaps of distribution per species were created by kernel density estimation in QGIS 3.16.5.

## 3. Results

### 3.1. Spring Survey in June 2017

3.1.1. Sightings

The planned 13 transects were covered in 5 days in the period 16–22 June with a realized total effort of 424 km. In total, 236 sightings of all three Black Sea cetaceans' species were recorded along the transect lines (Figure 3). On four occasions, species identification could not be made (Table 1). The average size of the groups varied between 1.31 for porpoises and 2 for bottlenose dolphins, with the common dolphin in between with 1.72.

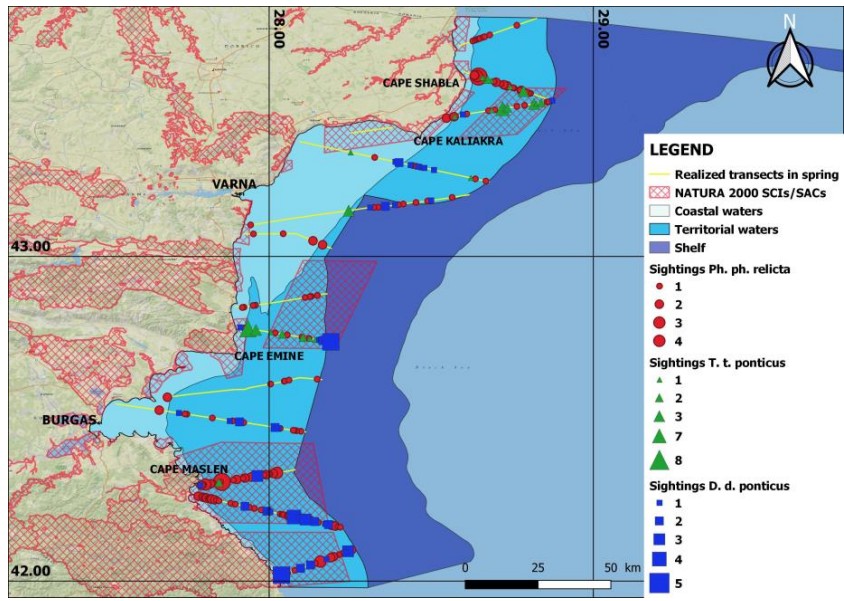

**Figure 3.** Realized transects and sighting of cetaceans in Bulgarian territorial waters in June 2017.

Porpoises were encountered mostly as solitary animals (73% of all sighting), followed by groups of two (23%). The largest encountered groups were of four porpoises, which was observed on only two occasions, while groups of three were seen four times. Encounter rate of porpoises was 0.39 sightings/km.

The largest observed group of common dolphins was of five and four, observed twice and only once, respectively. Groups of three were seen three times while solitary dolphins were dominant (56% of all sightings) followed by pairs (28%). Encounter rate of the common dolphins was 0.09 sightings/km.

**Table 1.** Number of sightings and individuals observed during the survey in June 2017.

| Species | Number of Sightings | Number of Individuals |
|---|---|---|
| *Tursiops truncatus ponticus* | 27 | 54 |
| *Delphinus delphis ponticus* | 39 | 67 |
| *Phocoena phocoena relicta* | 166 | 218 |
| *Unidentified cetacean* | 4 | 4 |
| Total | 236 | 343 |

The bottlenose dolphin was the species with the largest observed groups, with seven and eight seen once each. Groups of three were encountered on four occasions while solitary animals and pairs were observed most often: 56% and 22%, respectively. The encounter rate of bottlenose dolphins was 0.035 sightings/km.

3.1.2. Density, Abundance and Encounter Rate

The most reliable estimation for abundance of the Black Sea harbour porpoise was 9045 individuals (CV = 25.40%, 95% CI: 5301–15,433) using the hazard rate (HR) key function and simple-polynomial series extensions model for detection function in MCDS, with sea state as covariate (Table 2). To test whether applying global detection function would provide a more robust result, post-stratification by species was applied but obtained estimations' precision and accuracy were not any better than those for each dolphin species separately. The most reliable estimation for abundance of the Black Sea common dolphin was 2484 individuals (CV = 36.80%, 95% CI: 1192–5177) using the HR key function and simple-polynomial series extensions model of detection function in CDS. The same detection function model also provided the best estimation of abundance for Black Sea bottlenose dolphin: 1340 individuals (CV = 52.1%, 95% CI: 481–3735) (Figure 4).

**Table 2.** Encounter rates and estimations of abundance and density of groups and animals by species in June 2017.

| Parameter | Species | | |
|---|---|---|---|
| | *T. t. ponticus* | *P. p. relicta* | *D. d. ponticus* |
| Goodness-of-fit, CvM, *p*-value | >0.8 | >0.9 | >0.9 |
| Coefficient of variation (CV) % | 28.67 | 24.19 | 30.26 |
| 95% Confidence Interval (CI) | 0.01979–0.0622 | 0.235–0.663 | 0.049–0.177 |
| Effective strip width (ESW), m | 304.27 | 182.00 | 203.7 |
| Coefficient of variation (CV) % | 17.85 | 6.95 | 18.46 |
| 95% Confidence Interval (CI) | 211.29–438.16 | 158.70–208.72 | 140.58–295.16 |
| Estimate of density of clusters/groups *(DS)* | 0.105 | 1.083 | 0.227 |
| Coefficient of variation (CV) % | 49.34 | 25.17 | 35.45 |
| 95% Confidence Interval (CI) | 0.039–0.284 | 0.637–1.843 | 0.111–0.464 |
| Estimate of density of animals (*D*, individuals/km$^2$) | 0.211 | 1.423 | 0.391 |
| Coefficient of variation (CV) % | 52.15 | 25.40 | 36.84 |
| 95% Confidence Interval (CI) | 0.076–0.587 | 0.834–2.427 | 0.187–0.814 |
| Estimate of number of animals in the surveyed area (*N*) | 1340 | 9045 | 2484 |
| Coefficient of variation (CV) % | 52.15 | 25.40 | 36.84 |
| 95% Confidence Interval (CI) | 481–3735 | 5301–15,433 | 507–1827 |

*3.2. Autumn Survey in November–December 2017*

3.2.1. Sightings

The planned 12 transects were completed in the period 24 November to 22 December, with several interruptions caused by deteriorated weather conditions. During the survey in November–December, 123 sightings (Table 3) of all three Black Sea cetaceans' species were

recorded over 684 km of effort along the transect lines (Figure 5). Effort in the coastal shelf was 332.8 km while offshore it was 351.2 km. The average size of groups varied between 2.33 for common dolphins and 3.22 for porpoises with bottlenose dolphin in between with 2.375.

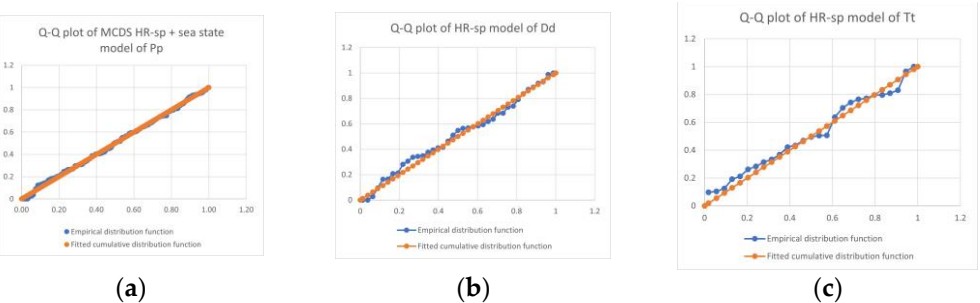

(**a**)           (**b**)           (**c**)

**Figure 4.** (**a**) Q–Q plot of the selected model of detection function for the harbour porpoise: MCDS hazard rate with simple polynomial extensions + sea state; (**b**) Q–Q plot of the selected model of detection function for the common dolphin: hazard rate with simple polynomial extensions; (**c**) Q–Q plot of the selected model of detection function for the bottlenose dolphin: hazard rate with simple polynomial extensions.

**Table 3.** Number of sightings and individuals observed during the survey in November–December 2017.

| Species | Number of Sightings | Number of Individuals |
| --- | --- | --- |
| *Tursiops truncatus ponticus* | 24 | 57 |
| *Delphinus delphis ponticus* | 18 | 42 |
| *Phocoena phocoena relicta* | 81 | 261 |
| Total | 123 | 360 |

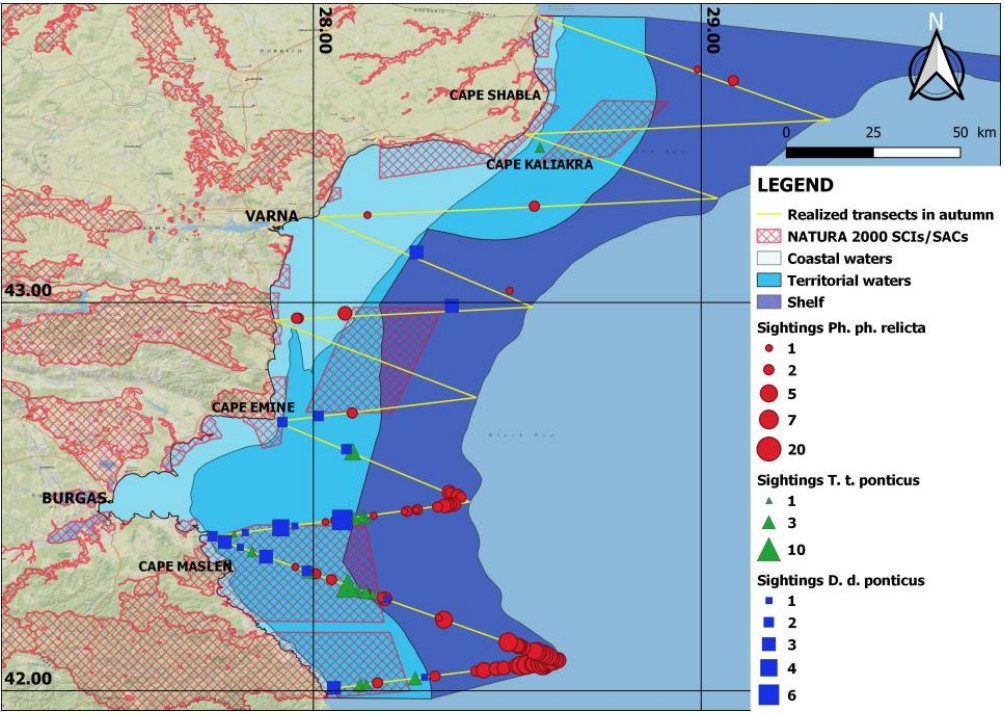

**Figure 5.** Realized transects and sightings of cetaceans in the Bulgarian shelf in November–December 2017.

Porpoises were encountered mostly in pairs (19% of all sightings), followed by groups of three and solitary animals (16% each). The largest encountered groups were of eight, nine and 20 porpoises observed in one occasion each, while groups of six and seven were seen three and four times, respectively. Groups of four (7%) and five (10%) were relatively more frequent. The encounter rate of porpoises was the highest at 0.118 sightings/km.

Bottlenose dolphins were the second most encountered species during the autumn. Solitary dolphins (33% of all sightings) were seen most often followed by groups of two and three (29% each). The largest observed groups were of four and ten dolphins, which were each observed only once. The encounter rate of bottlenose dolphins was 0.035 sightings/km.

The largest observed groups of common dolphins were of six and four, observed only once each. Pairs were dominant (33% of all sightings) while groups of three and solitary dolphins were seen on five occasions (28%) each. The encounter rate of common dolphins was 0.026 sightings/km.

### 3.2.2. Density, Abundance and Encounter Rate

Population parameters for all three cetaceans (abundance, density, group sizes) were estimated in Distance 7.0 software using CDS. For the harbour porpoise, the most accurate results were obtained by the hazard rate key function with simple polynomial series extensions model for detection function. For the two dolphin species, the half-normal key function with cosine extensions model of detection function calculated the most precise estimations (Figure 6).

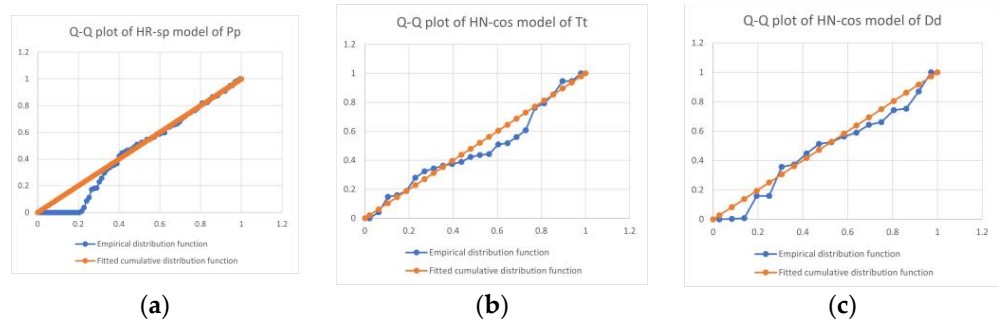

**Figure 6.** (**a**) Q–Q plot of the selected model of detection function for the harbour porpoise: hazard rate with simple polynomial extensions; (**b**) Q–Q plot of the selected model of detection function for the bottlenose dolphin: half normal with cosine extensions; (**c**) Q–Q plot of the selected model of detection function for the common dolphin: half-normal with cosine extensions.

The results showed that there was a significant difference (almost eight times higher) in the densities of porpoises in the coastal and offshore shelf: 0.136 ind./km$^2$ (CV = 35.82%) and 1.065 ind./km$^2$ (CV = 43.4%). Bottlenose dolphins showed a four times higher density in the coastal versus offshore shelf: 0.187 ind./km$^2$ (CV = 52.13%) and 0.042 ind./km$^2$ (CV = 71.07%). Common dolphin densities were higher also by factor of 4 in inshore waters: 0.138 (CV = 48.59%) and 0.031 ind./km$^2$ (CV = 57.63%). The results of encounter rates and estimates of densities of groups, densities and abundance of animals by stratum and for the global shelf are shown in Table 4.

### 3.3. Distribution and Hotspots in Spring and Autumn

The survey in June revealed two hotspots in the spatial distribution of porpoises in Bulgarian territorial waters: the first one in the northern part opposite Cape Shabla and second in the southern part from Cape Maslen and further south into Strandzha SCI BG0001007. Both these hotspots were at depths between 40 and 70 m (Figure 7). Common dolphins' main hotspot was in the south, slightly south and east (within SCI Ropotamo BG0001001 and Strandzha BG0001007) when compared to the porpoises' southern hotspot, with a secondary one south of Cape Kaliakra and smaller one on the offshore edge of the

territorial waters opposite Cape Emine. The southern hotspot was between 50 and 80 m depth while the northern one was encompassing depths from 30 to 70 m. A secondary hotspot in the central part opposite Cape Emine was at 70 m depth (Figure 8). Bottlenose dolphins had a major hotspot overlapping with porpoises off Cape Shabla (at SCI Kompleks Kaliakra BG0000573) at depths from 40 to 70 m, with a secondary smaller one just off Cape Emine (SCI Emona BG0001501) in the central part at a slightly shallower shelf between 40 and 50 m (Figure 9).

**Table 4.** Estimated encounter rate, density and abundance of groups and individuals by species in November–December 2017.

| Parameter | Species | | |
|---|---|---|---|
| | *T. t. ponticus* | *P. p. relicta* | *D. d. ponticus* |
| Number of observations (individuals or groups)–coastal/offshore | 19/5 | 14/67 | 15/3 |
| Goodness-of-fit, CvM, *p*-value | >0.5 | >0.05 | >0.6 |
| Encounter rate (ER; *n/L*)–coastal shelf | 0.057 | 0.042 | 0.045 |
| Coefficient of variation (CV) % | 46.3 | 31.98 | 43.97 |
| 95% Confidence Interval (CI) | 0.022–0.022 | 0.021–0.084 | 0.018–0.114 |
| Encounter rate (ER; *n/L*)–offshore shelf | 0.014 | 0.191 | 0.009 |
| Coefficient of variation (CV) % | 67.61 | 46.16 | 48.69 |
| 95% Confidence Interval (CI) | 0.004–0.05 | 0.073–0.502 | 0.003–0.024 |
| Estimate of density of groups *(DS)*–coastal | 0.077 | 0.073 | 0.071 |
| Coefficient of variation (CV) % | 48.32 | 34.52 | 45.98 |
| 95% Confidence Interval (CI) | 0.029–0.208 | 0.036–0.149 | 0.028–0.183 |
| Estimate of density of groups *(DS)*–offshore | 0.042 | 0.331 | 0.013 |
| Coefficient of variation (CV) % | 69 | 47.96 | 50.52 |
| 95% Confidence Interval (CI) | 0.005–0.075 | 0.124–0.887 | 0.005–0.038 |
| Estimate of density of animals (*D*, individuals/km$^2$)–coastal | 0.187 | 0.136 | 0.138 |
| Coefficient of variation (CV) % | 52.13 | 35.82 | 48.59 |
| 95% Confidence Interval (CI) | 0.067–0.526 | 0.0652–0.282 | 0.052–0.366 |
| Estimate of density of animals (*D*, individuals/km$^2$)–offshore | 0.042 | 1.065 | 0.031 |
| Coefficient of variation (CV), % | 71.07 | 48.82 | 58.04 |
| 95% Confidence Interval (CI) | 0.011–0.168 | 0.394–2.877 | 0.0099–0.1 |
| Estimate of number of animals in the coastal shelf (*N*) | 1191 | 863 | 878 |
| Coefficient of variation (CV), % | 52.13 | 35.82 | 48.59 |
| 95% Confidence Interval (CI) | 424–3346 | 415–1795 | 331–2329 |
| Estimate of number of animals in the offshore shelf (*N*) | 243 | 6106 | 180 |
| Coefficient of variation (CV), % | 71.07 | 48.82 | 58.04 |
| 95% Confidence Interval (CI) | 61–965 | 2261–16,488 | 56–574 |
| Estimate of density of animals (*D*, individuals/km$^2$) in the global shelf | 0.119 | 0.576 | 0.088 |
| Coefficient of variation (CV), % | 45.52 | 43.43 | 42.13 |
| 95% Confidence Interval (CI) | 0.048–0.293 | 0.237–1.401 | 0.038–0.204 |
| Estimate of number of animals in the global shelf (*N*) | 1434 | 6969 | 1058 |
| Coefficient of variation (CV), % | 45.52 | 43.43 | 42.13 |
| 95% Confidence Interval (CI) | 581–3537 | 2867–16,938 | 454–2466 |

In the autumn, the main hotspot of common dolphins showed a slight northern shift to depths between 30 and 50 m in the southern part, but still being within the coastal waters and SCI Ropotamo BG0001001. Bottlenose dolphins' only hotspot in the autumn was in the south at deeper waters (50 to 70 m) just at the offshore edge of territorial waters opposite Cape Maslen nos (SCI Ropotamo BG0001001). The most distinctive shift was demonstrated by harbour porpoises. The presence of harbour porpoises in the coastal waters was rare with more than 80% of encounters, including the largest observed groups, being in the

offshore shelf. Only one hotspot was detected in the autumn at the extreme southeastern border of the shelf, where the deepest waters are between 90 and 100 m.

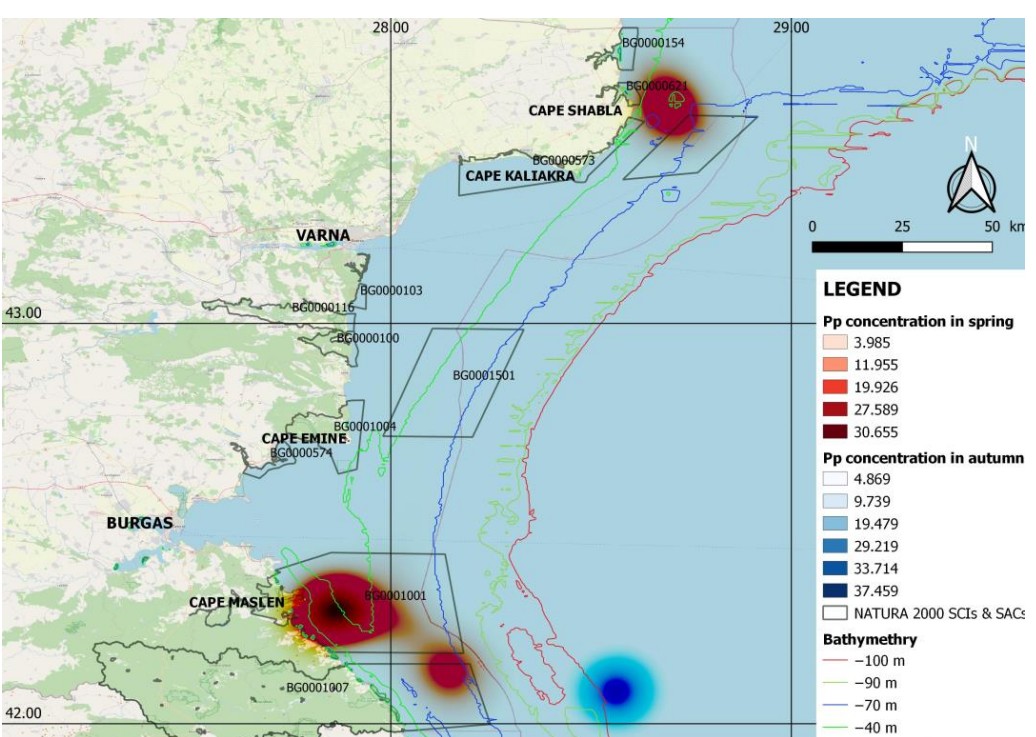

**Figure 7.** Heatmap of the Black Sea harbour porpoise in the Bulgarian shelf in the spring and autumn of 2017.

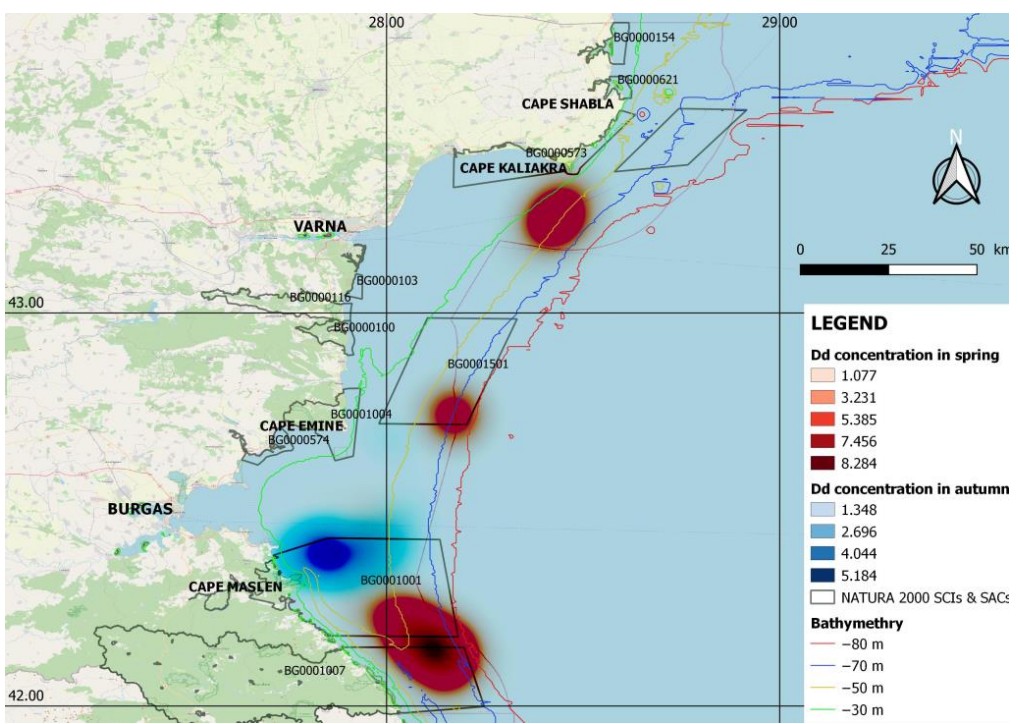

**Figure 8.** Heatmap of the Black Sea common dolphin in the Bulgarian shelf in the spring and autumn of 2017.

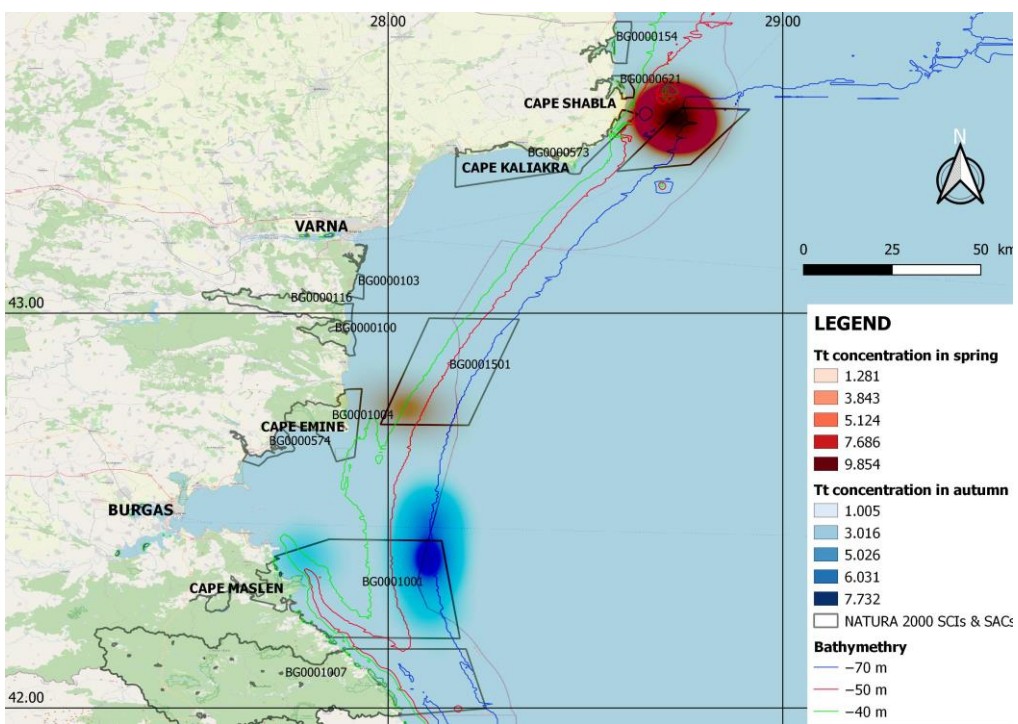

**Figure 9.** Heatmap of the Black Sea bottlenose dolphin in the Bulgarian shelf in the spring and autumn of 2017.

## 4. Discussion

Abundance, density and distribution, including their spatial and temporal variation, are the basic but among the most important population parameters of cetaceans used for the assessment of their conservation status. Regular surveys over long periods are needed to detect trends in these parameters. All three species are highly mobile top predators in the Black Sea ecosystem and variations in their distribution pattern is highly dependent on prey distribution [22]. The results of the conducted surveys in 2017 provide robust baseline estimates for density and abundance of cetaceans in the Bulgarian shelf over two seasons: late spring and late autumn in the particular year. Obtained estimations are compared with similar surveys conducted in the Black Sea coastal and shelf waters over the last 20 years.

### 4.1. Black Sea Harbour Porpoise

The Black Sea harbour porpoise is the most abundant species in both surveyed seasons with a well-defined shift from coastal to offshore waters. The overall encounter rate in the spring was more than three times higher than in the autumn, while if we compare only the coastal shelf, the decrease is ninefold. Densities in coastal waters in autumn decrease by factor of more than ten compared to June. At the same time, the shift to the offshore waters of the shelf, most visible in the southern part, is underlined by the fact that the density in offshore waters in autumn is just 33% lower compared to coastal waters in the spring. That shift coincides with well-documented migration of the porpoises within the Black Sea from west/northwest to south/southeast (Turkish and Georgian water) in winter [23]. The average group size also increased between spring and autumn indicating migration and typical winter concentrations described for that species [3]. Available data from previous surveys in coastal waters around the Black Sea is limited, but a comparison is still worthwhile. In the summer of 2013, an international vessel survey deploying the same methodology was conducted covering territorial waters of Bulgaria, Romania and part of the Ukraine (west of Cape Tarkhankut, Crimea). Results of this survey [6] are the best baseline for a comparison of obtained estimations in our surveys as it had the same study area as the survey in June, despite a slightly smaller effort coverage: 347.5 km versus 424 km

in our case. The estimated density of the Black Sea harbour porpoise in Bulgarian territorial waters in July 2013 was 0.144 ind./km$^2$, which is almost ten times lower than in June 2017; however, this is a close to estimation for coastal waters in autumn 2017 when that parameter was estimated to 0.136 ind./km$^2$. The survey in June 2017 in adjacent southern territorial waters of Romania in the Black Sea detected the highest reported density of porpoises so far: 5.359 ind./km$^2$ [24], though in a much smaller study area of 1063 km$^2$. That, like all other numbers cited here, shows the relative densities and abundances valid for respective study areas that vary in size and environmental features considerably. One factor that defines estimations is the size of surveyed area, as surveying smaller but denser areas produces higher values. That is well shown in the results of pilot surveys conducted in two Bulgarian marine protected areas in spring of 2016 when densities of porpoises were 0.871 ind./km$^2$ in Strandzha SCI [14] and 1.867 ind./km$^2$ in Ropotamo SCI [15]. A similar high density as in Romanian waters is reported only for coastal waters of southeastern Crimea in April 2011 at 4.86 ind./km$^2$, and for an even smaller study area of only 105 km$^2$ [25]. In autumn 2015, a pilot survey [13] in the central Bulgarian shelf covering 2540 km$^2$ provided a density estimation of 0.15 ind./km$^2$, which is almost four times lower than the survey in autumn 2017. The most probable reason for that is the fact that the surveyed area in 2015 covered only the central shelf thus missing the hotspot area in the southern offshore shelf detected in autumn 2017.

### 4.2. Black Sea Common Dolphin

The Black Sea common dolphin is the second most abundant species in the spring, while in the autumn, its abundance is the lowest of all three species. The overall encounter rate in spring is more than three times higher than in autumn. If we compare only the coastal shelf, the decrease is two-fold, while in the offshore shelf it is 10-fold. That result is contradictory to the usual preferences ascribed to common dolphins [4,22], but on the other hand, should be related to prey availability during the periods of the surveys and the fact these covered only the shelf. Densities in coastal waters in autumn decrease by factor of almost three compared to June, while in the offshore shelf, that factor is above 12. The average group size of observed common dolphins were also low, varying between 1.72 and 2.33, which is much lower than the typically described groups of tens, hundreds and even thousands in the past [4,9,26]. Comparing obtained estimations with previous surveys reveal that in spring 2017, the density in coastal waters was almost two-fold lower than in July 2013 when it was 0.718 ind./km$^2$ [6]. On the other hand, it was more than two-fold higher than in adjacent Romanian waters in June 2017, when that parameter was estimated as 0.153 ind./km$^2$ [24]. Despite a decreased density observed in autumn, it must be noted that during the pilot survey, conducted in November 2015 in the central Bulgarian shelf, that species was not detected at all [13]. In the current survey, four sightings were detected at that part of the shelf, but the highest observed density was in southern coastal waters. The overall low density of common dolphins in the shelf in both seasons when compared to the harbour porpoise well concurs with historical data on the distribution of common dolphins, describing depth as the main factor [26].

### 4.3. Black Sea Bottlenose Dolphin

The Black Sea bottlenose dolphin is the least abundant species in the spring, while in the autumn, its abundance is the second highest of all three species. The overall encounter rate in the spring and autumn is identical. If we compare only the coastal shelf, an increase of 60% is observed while in the offshore shelf, the decrease is 2.5-fold. The density in coastal waters is almost constant between spring and autumn, while in offshore waters, it is almost five-fold lower in the autumn. These results coincide with the preferred habitat by that species in coastal and shallower waters described in recent research in the Black Sea [12,24,27,28]. Comparing obtained estimations with previous surveys revealed that in spring 2017, the density in coastal waters was more than three-fold lower than in July 2013 when it was 0.696 ind./km$^2$ [6]. It was also two-fold lower than in adjacent Romanian

waters in June 2017 when that parameter was estimated as 0.424 ind./km$^2$ [24]. The density in the autumn was lower not only in comparison to spring, but also to the estimation in the central Bulgarian shelf [13] in autumn 2015, which was 0.323 ind./km$^2$, a value almost threefold higher than in autumn 2017. The lowest density of bottlenose dolphins was detected during the pilot surveys in the Bulgarian marine protected area Strandzha SCI in spring 2016: 0.107 ind./km$^2$ in May [14]. The surveys show that bottlenose dolphins make a significant shift to the south in Bulgarian shelf between spring and autumn and show a marked preference for shallower waters.

## 5. Conclusions

All three Black Sea cetacean species have been detected during the conducted surveys in the spring and autumn of 2017 of the Bulgarian shelf. The Black Sea harbour porpoise was the most abundant species in both seasons with a well-defined shift from the coastal to offshore shelf between spring and autumn. That shift could be justified by both prey distribution and seasonal migration hinted by larger groups observed in the autumn. The Black Sea common dolphin has shown a distribution pattern over the shelf that contradicts the basin-wide preferences that are mainly driven by depth and related prey distribution. This can be attributed to the finer resolution of the study area while overall low density of that species can further support its preference for deep pelagic waters. The Black Sea bottlenose dolphin is the least abundant species concurring with existent data for it at basin level [6,29]. Detected hotspots in the spring overlap to a great extent with existing marine sites of community interest (SCI) and special areas of conservation (SAC) in Bulgarian Black Sea waters that are part of the pan-European ecological network NATURA 2000. These sites, declared under the EU Habitats Directive 92/43/EEC, include both harbour porpoises and bottlenose dolphins among their conservation objectives. The spring hotspots of porpoises overlap with the following SACs: BG0000621 Shabla-Ezerets and BG0000573 Kompleks Kaliaka in the north and SCI BG0001007 Strandzha and SAC BG0001001 Ropotamo in the south. The only winter porpoise hotspot is outside of existing NAURA 2000 coverage. The two spring hotspots of bottlenose dolphins overlap with the following SACs: BG0000621 Shabla-Ezerets and BG0000573 Kompleks Kaliaka in the north and SCI BG0001501 Emona in the central shelf. The only winter hotspot overlaps with BG 0001001 Ropotamo in the south.

Data was used during a workshop for identification of Important Marine Mammal Areas (IMMAs) in the Black Sea in 2021. All detected hotspots have been encompassed by the identification of three IMMAs that overlap with our study area: Kaliakra to Danube Delta, Emona and Western Black Sea "https://www.marinemammalhabitat.org/imma-eatlas/ accessed on 12 December 2022)".

The presented results provide new data on the seasonal distribution and variation of abundances of three cetacean species of high conservation concern. As mentioned above, these are subject to numerous conservation measures including international agreements and two directives of the European Union. As a member state, Bulgaria is required to ensure effective conservation of these species and to report on their conservation status every 6 years. Designated SACs are a positive step in that direction, but are not sufficient, as effective conservation requires active management including restriction of harmful human activities. The Black Sea is an enclosed continental sea whose ecosystem has suffered a major degradation over the last century. Currently, only two of the six riparian countries are members of the EU that is setting high standards in biodiversity conservation; however, for mobile species, such as cetaceans, borders do not exist and a holistic approach with participation of all coastal states' governments is the only way to effectively ensure long-term survival of the fragile Black Sea ecosystem and all its species. Moreover, for proper assessment of conservation status and trend in abundance of these mobile species, basin-wide surveys are needed at intervals of 5–6 years. Local surveys, such as those described here, are supplementary and should be conducted as often as possible to provide early alert for changes in distribution and abundance.

**Author Contributions:** Conceptualization, D.P.; methodology, D.P. and M.P.; software, D.P.; validation, M.P. and H.D.; formal analysis, D.P.; investigation, G.M., R.B., D.P. and M.P.; resources, D.P. and M.P.; data curation, D.P. and M.P.; writing—original draft preparation, D.P.; writing—review and editing, G.M., M.P., R.B. and H.D.; visualization, D.P.; supervision, M.P. and H.D.; project administration, D.P. and M.P.; funding acquisition, D.P. and M.P. All authors have read and agreed to the published version of the manuscript.

**Funding:** This research was funded by OceanCare, Switzerland and through Agreement between MOEW and IO-BAS for the fulfilment of the national monitoring requirements of Bulgaria to the EU Marine Strategy Framework Directive for 2017.

**Institutional Review Board Statement:** Not applicable.

**Data Availability Statement:** The data presented in this study are available on request from the corresponding author. The data are not publicly available due to not being funded by public funding.

**Acknowledgments:** Authors wish to thank funders of that research–OceanCare and MOEW. Special thanks to Green Balkans NGO's volunteers Antonia Miteva, Nikolay Davidkov, Polina Hristova and Teodora Ilieva who helped during data collection as well as shipmasters of vessels–Stoyan Georgiev and Dincher Yusein–who ensured smooth performance of surveys.

**Conflicts of Interest:** The authors declare no conflict of interest. The funders had no role in the design of the study; in the collection, analyses, or interpretation of data; in the writing of the manuscript; or in the decision to publish the results.

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
