# Peer review of "Seasonal Abundance, Density and Distribution of Cetaceans in the Bulgarian Black Sea Shelf in 2017"

_diversity, doi:10.3390/d15020229_

Round 1

Reviewer 1 Report

Check if Fig. 1, Fig. 2 and Fig. 3 are cited in text.

Page 10, Line 281 - table 4 should be Table 4.

Page 11 , Line 289 - Cape Masle nos should be north?

Page 15, Line 426 - Komlex should be Komplex.

Reference should be arranged in alphabetical  and in chronological order. See [18] - Buckland et al., 2001 is after [16]; Panayton et al., 2020  is after [15] - Popov et al., 2021.

A personal suggestion: [8] ACCOBAMS, 2021 - first should be name of author(s), title of article/report. In: ACCOBAMS ....  

Eremeev and Zuyev (2007) should be before Nikolov, 1963.   (See and other possible improvement)

Author Response

Hello,

Thanks a lot for your comments. All these are revised: figures 1-3 are added at appropriate sections in text. 

Concerning references, all these are listed according Diversity Journal's Instruction for Authors:
References: References must be numbered in order of appearance in the text (including table captions and figure legends) and listed individually at the end of the manuscript.

Then numbering is according appearance in the text and thus no way to be in alphabetical or chronological order.

ACCOBAMS report is cited according recommended way of citation in the report.

Revised manuscript is attached.

Kind regards,

Dimitar Popov

Reviewer 2 Report

I have revised the paper on cetacean in the bulgarian area dealing with two line transect distance sampling surveys from boat. The data are interesting in the MSFD perspective and deserve publication. Nevertheless I am suggesting some revision in the text and in the structure of the paper before publication 

To this extent I attach here the pdf with some comments and suggestions for the needed improvement of the manuscript. 

In general there are a lot of redundant information; some are presented in both the text and in the tables...I suggest to identify just a single way to present data also in order to shorten the manuscript and improve legibility. 

Moreover, some data data needs to be moved in the right section. 

As the results are concerned I would suggest to make it more simple in the description of the parameters used; in the table I would suggest to delete the SE since CV% and 95%CI are presented as measure of the uncertainty of each value. A single table comparing just densities and abundances for each species estimates for the two surveys will help in the results comparison. 

The locations indicated in the text (Cape Shabla, Maslen, etc) are not reported in the maps and readers can not well understand the distribution of the animals in relationship to those area, expecially in the case of the SACs, SCI and IMMAs. As these conservation areas is concerned it will be usefull to have their boundaries in the maps. The Kernel distribution in the maps is not well represented since just two values/colours are in the legends but much more colours are presented in the maps. 

Author Response

Hello,

Thank you very much for valuable comments and proposed revisions. All these are addressed in the revised manuscript that is attached.

Figures are revised with added locations and better colours. These are included in revised manuscript.

Best regards,

Dimitar Popov